# National Survey: How Do We Approach the Patient at Risk of Clinical Deterioration outside the ICU in the Spanish Context?

**DOI:** 10.3390/ijerph191912627

**Published:** 2022-10-03

**Authors:** Álvaro Clemente Vivancos, Esther León Castelao, Álvaro Castellanos Ortega, Maria Bodi Saera, Federico Gordo Vidal, Maria Cruz Martin Delgado, Cristina Jorge-Soto, Felipe Fernandez Mendez, Jose Carlos Igeño Cano, Josep Trenado Alvarez, Jesus Caballero Lopez, Manuel Jose Parraga Ramirez

**Affiliations:** 1Health Sciences Doctoral Program, Universidad Católica de Murcia (UCAM), 30107 Murcia, Spain; 2Advanced Nursing Practice, Hospital del Mar, 08003 Barcelona, Spain; 3IMIM (Hospital del Mar Medical Research Institute), 08003 Barcelona, Spain; 4Simulation Laboratory, School of Medicine and Health Sciences, 08036 Barcelona, Spain; 5Clinical Simulation Lab, University of Barcelona, 08036 Barcelona, Spain; 6Intensive Care Unit Medical Director, University Hospital La Fe, 46026 Valencia, Spain; 7Associate Lecturer, University of Valencia, 46010 Valencia, Spain; 8Intensive Care Unit, University Hospital Joan XIII, 43005 Tarragona, Spain; 9Pere I Virgili Health Research Institute, Rovira I Virgili University, 43003 Tarragona, Spain; 10Center for Biomedical Research in Respiratory Diseases Network (CIEBERES), Carlos III Health Institute, 28029 Madrid, Spain; 11Intensive Care Unit, University Hospital of Henares, 28822 Madrid, Spain; 12Critical Pathology Research Group, Francisco de Vitoria University, 28223 Madrid, Spain; 13Intensive Care Unit, Hospital 12th of October, 28041 Madrid, Spain; 14Facultad de Medicina, Francisco de Vitoria University, 28223 Madrid, Spain; 15CLINURSID Research Group, Psychiatry, Radiology, Public Health, Nursing and Medicine Department, Universidad de Santiago de Compostela, 15705 Galicia, Spain; 16Simulation and Intensive Care Unit of Santiago (SICRUS) Research Group, Health Research Institute of Santiago, University Hospital of Santiago de Compostela-CHUS, 15705 Santiago de Compostela, Spain; 17Faculty of Nursing, Universidade de Santiago de Compostela, 15705 Santiago de Compostela, Spain; 18School of Nursing, Universidade de Vigo, 36310 Pontevedra, Spain; 19REMOSS Research Group, Universidade de Vigo, 36310 Pontevedra, Spain; 20Intensive Care Unit, San Juan de Dios Hospital, 14012 Cordoba, Spain; 21Intensive Care and High Dependency Unit, Mutua Terrassa Hospital, 08221 Terrasa, Spain; 22Department of Medicine, University of Barcelona, 08036 Barcelona, Spain; 23Intensive Care Unit, University Hospital Arnau de Vilanova, 25198 Lleida, Spain; 24IRBLleida, 25198 Lleida, Spain; 25Intensive Care Unit, JM Morales Meseguer, 30008 Murcia, Spain; 26Simulation and Clinical Skills Director, UCAM, 30107 Murcia, Spain; 27Medical Degree Direction Team, UCAM, 30107 Murcia, Spain

**Keywords:** rapid response team, hospital medical emergency team, early warning score

## Abstract

Background: Anticipating and avoiding preventable intrahospital cardiac arrest and clinical deterioration are important priorities for international healthcare systems and institutions. One of the internationally followed strategies to improve this matter is the introduction of the Rapid Response Systems (RRS). Although there is vast evidence from the international community, the evidence reported in a Spanish context is scarce. Methods: A nationwide cross-sectional research consisting of a voluntary 31-question online survey was performed. The Spanish Society of Intensive, Critical and Coronary Care Medicine (SEMICYUC) supported the research. Results: We received 62 fully completed surveys distributed within 13 of the 17 regions and two autonomous cities of Spain. Thirty-two of the participants had an established Rapid Response Team (RRT). Common frequency on measuring vital signs was at least once per shift but other frequencies were contemplated (48.4%), usually based on professional criteria (69.4%), as only 12 (19.4%) centers used Early Warning Scores (EWS) or automated alarms on abnormal parameters. In the sample, doctors, nurses (55%), and other healthcare professionals (39%) could activate the RRT via telephone, but only 11.3% of the sample enacted this at early signs of deterioration. The responders on the RRT are the Intensive Care Unit (ICU), doctors, and nurses, who are available 24/7 most of the time. Concerning the education and training of general ward staff and RRT members, this varies from basic to advanced and specific-specialized level, simulating a growing educational methodology among participants. A great number of participants have emergency resuscitation equipment (drugs, airway adjuncts, and defibrillators) in their general wards. In terms of quality improvement, only half of the sample registered RRT activity indicators. In terms of the use of communication and teamwork techniques, the most used is clinical debriefing in 29 centers. Conclusions: In terms of the concept of RRS, we found in our context that we are in the early stages of the establishment process, as it is not yet a generalized concept in most of our hospitals. The centers that have it are in still in the process of maturing the system and adapting themselves to our context.

## 1. Introduction

Today, there is an aging population with more comorbidities and long-term pharmacological treatments that frequently undergo invasive procedures and interventions. This situation implies an increased risk, associated with the exacerbation of their chronic pathological processes, which implies an increase in the need for Intensive Care Unit (ICU) beds, an increase in morbidity and mortality, as well as greater hospital follow-up, which makes their care a much more complex process for healthcare organizations [1]. 

In the 1990s, countries such as the United States of America (USA), Australia, and the United Kingdom (UK) wondered why patients admitted to the ICU after resuscitation maneuvers from hospital wards had higher mortality and a worse prognosis than those admitted to the emergency room or operating room [2,3]. These events were thought to have a “sudden and unexpected” appearance, however, upon investigating, they concluded that there is a risk of these critical events, and they are normally preceded by measurable alterations of pathophysiological deterioration (alterations in vital signs and analytical values), in some cases present up to more than 24 h before undergoing a cardiac arrest (CA) [4,5].

The delay in the recognition and early treatment of these patients leads to an increase in mortality, average stay (hospital and ICU), unscheduled or urgent complications, and admissions to the aforementioned unit [6,7].

New approaches are designed in these countries, focusing on the complexity of care (level of care) that each patient requires individually, regardless of the location of the patient around the hospital [8,9]. The process of the critically ill patient highlights the need to scale these approaches at a system level, creating Rapid Response Systems (RRS) as a safety net for patients at risk of clinical deterioration, as well as along the critical illness process [10].

These RRS are designed based on four limbs: (a) The afferent limb (how the system is activated), (b) the efferent limb (the response of the system to activation), (c) the administration limb (provides material, human, and training resources to the system), and (d) the limb for continuous healthcare improvement and patient safety (responsible for recording, measuring, and analyzing data, as well as providing feedback to the institution) [10].

Minimal evidence and experiences have been published in a Spanish context regarding RRS, with most of these experiences coming from pre-post intervention research such as the ones from University Hospital Marqués de Valdecilla [11] or the ones from University Hospital de Henares [12]. To assess variations and specifics on how hospitals have designed and built up their RRS in a Spanish context, we designed a 31 item questionnaire which was supported and distributed nationwide by the Spanish Society of Intensive-Critical Care Medicine and Coronary Care Units (SEMICYUC).

### Aim

The aim of this research is to describe and analyze the characteristics of the organizations and the process of care for patients at risk of clinical and/or serious deterioration outside intensive care units in the Spanish context. 

## 2. Materials and Methods

### 2.1. Research Design

A cross-sectional study with a descriptive design. Supported by SEMICYUC, purposive sampling was used to choose from its list of hospitals with Intensive Care Units service. 

### 2.2. Inclusion and Exclusion Criteria

The inclusion criteria used in our study were hospitals in the SEMICYUC list of centers with an ICU that agreed to participate in the research. The exclusion criterion was having an incomplete online survey.

### 2.3. Instruments

A specific survey questionnaire was developed for this research, containing 31 items based on the four limbs of the RRS: Afferent, Efferent, Administrative, and Quality improvement. The creating and validating process of this questionnaire underwent the following steps: (1) Systematic review; (2) International variables screening; (3) Questionnaire first version; (4) Cultural adaptation; (5) Questionnaire second version; (6) First Expert’s review (three nationwide ICU background healthcare professionals); (7) Questionnaire third version; (8) Second Expert´s review (six nationwide ICU background healthcare professionals from SEMICYUC); (9) Questionnaire fourth version, pilot test of the questionnaire, research information sheet and informed consent (performed by four nationwide ICU Medical directors responsible for RRS); (10) Questionnaire fifth and definitive version. 

### 2.4. Procedure

The study was conducted over a seven-month period, from July 2020 to January 2021. The survey was distributed via email by SEMICYUC nationwide ICU Medical directors’ lists. The email contained: a copy of the information sheet, a copy of the informed consent, and a link to the questionnaire allocated on the online secure server of the principal investigator University. Reminder emails were sent 3 months after the first trial. 

Although this research did not directly involve or affected patients and consistent with the Helsinki declaration, the research protocol was approved by the Ethics Committee of Universidad Católica de Murcia (UCAM) (Approval number CE062006; 26 July 2020 approved research protocol). 

### 2.5. Variables

Variables used in our study, obtained from the questionnaire, were divided into two parts.

#### 2.5.1. Socio-Demographic Variables

Region, presence of a Rapid Response Team (RRT), and RRT year of establishment.

#### 2.5.2. Afferent Limb

Frequency on measuring vital signs, vital signs measured, vital signs registration, discriminating abnormality, use of Early Warning Scores (EWS), automated alarms on abnormal parameters, professional allowed to activate Rapid Response Team (RRT), method for activating RRT, moment of activation, reasons for activating RRT, number of digits of phone number to activate RRT, first professional contacted when a patient deteriorates. 

#### 2.5.3. Efferent Limb

RRT common responders, RRT common members, RRT availability, RRT leadership, RRT role distribution.

#### 2.5.4. Administrative Limb

Resuscitation guidelines, ward nurses training, RRT members training, simulation training and facilities, resuscitation material presence and allocation, defibrillators presence and allocation.

#### 2.5.5. Quality Improvement Limb

Information and data collected by RRTs, use of structured communication tools (use of briefing, huddle, and clinical debriefing). 

### 2.6. Statistical Analysis

Descriptive statistics were calculated to obtain the frequencies and proportions that would enable a description of sociodemographic characteristics and limb variables. Statistical analysis was performed with SPSS statistical software (version 24; SPSS Inc., Chicago, IL, USA).

## 3. Results

### 3.1. Introduction: Demographics and RRS

Responses were received from 62 hospitals (20.6%) of the 300 centers with ICU registered at the SEMICYUC website. Responses were distributed by 13 of the 17 counties in Spain. We did not obtain any answers from the following four regions: Castilla la Mancha, Comunidad Foral de Navarra, and Extremadura y La Rioja. We also did not obtain any answers from the cities of Ceuta and Melilla.

Of the 62 responders, 32 (51.6%) have a dedicated team to attend to the patient who suffers from clinical deterioration. These RRTs have been implemented over the last ten years, and nine (14.5%) of them were implemented within the last two years.

### 3.2. The Afferent Limb: How the Risk of Deterioration Is Detected, and the System Activated 

As shown in Table 1, in terms of frequency of measuring vital signs, most of the participants answered that they record them at least once per shift contemplating other frequencies (48.4%) or only once per shift (43.5%). Hemodynamic-related vital signs are most recorded, while respiratory ones such as fraction of inspired oxygen (FiO_2_) (41.9%) and respiratory rate (50%) are less measured and recorded. At the time of registering vital signs, most of the samples (66.1%) register them on paper, and then introduce them manually into the electronic medical history. Regarding discriminating abnormality within vital signs, only 12 (10.4%) use EWS to interpret and respond to clinical deterioration. For example, at 14 (22%) of these centers, their vital signs measurement is compatible with the standardized National Early Warning Score 2 (NEWS2). 

Table 2 represent how the RRS is activated. We found that 55% of the sample doctors and nurses can activate the RRT, and in 39%, all healthcare professionals are allowed to do it. The most common methods for activating the RRS are via unidirectional (27.4%) or bidirectional (80.7%) telephone calls, where we found 56 different types of numbers to call. It is important to remark that in 11.3% of the sample, RRTs were activated within early signs of deterioration instead of when patients already had established signs of instability (80.6%). Apart from changes in vital signs (85.5%) or clinical signs of deterioration (82.3%), another important reason stated to call the RRT is “Professional concern” or “being worried” about the patient (56.5%).

### 3.3. The Efferent Limb: Once Risk Is Detected, How the System Reponds to It

As described in Table 3, regarding the efferent limb, at 17 (15%) of the hospitals, the responder was the on-call clinician–intensivist, whereas in the other 45 (85%) centers, responders were from one team (74%) or more than one team (26%) attending the deteriorating patients. Different names were given to these teams: Cardiopulmonary Resuscitation team (30%), Extended ICU (16%), or Advanced Life Support team (11%), among others.

As specified in Table 3, 57 (92%) of the hospitals had a response team or clinician available for 24 h/7 days, mostly depending on the Intensive Care Service (97%), Emergency department (1.5%), and mixed service (Anesthesia, Cardiology, and Internal Medicine) (1.5%). The most common professionals attending with RRTs are ICU consultant (95.5%), ICU trainee (53.2%), ICU nurse (24.2%), and ICU RRT full time dedicated nurse (3.2%). In the present research, we found in several participants that leadership (92.8%) and roles distribution (67.7%) were preestablished.

### 3.4. The Administrative Limb: Provides Human, Educational, and Material Resources to the System

#### 3.4.1. Educational Resources

Regarding the guidelines used in terms of cardiopulmonary resuscitation (CPR) algorithms, the most frequently used are the ones established by European Resuscitation Council (ERC) [13] (71%). Other guidelines are used, like American Heart Association (AHA) [14], relying the decision on which to use on the units (11.3%) or the team leader (12.9%).

As shown in Table 4, general ward staffing receives different types of training, from knowledge and skills on who and how-to call-in case of a clinical emergency (40.3%), through Basic Life Support (BLS) (54.8%) and up to Immediate (38.7%)/Advanced Life Support (ALS) (38.7%). In terms of RRTs members’ training, it ranged from no standardized training plan (17.7%), through Advanced Life Support (41.9%), to other specific training and experience (72.6%) (i.e., intensive/anesthetics care skills).

In our study, as described in Table 4, simulation is present in most of the participant centers: up 51.6% have simulation activities, and between 11.6% and 11.3% have, respectively, a dedicated simulation space or unit center with a dedicated team.

#### 3.4.2. Material Resources

We asked the participants about different types and allocations of specific resuscitation material. In terms of resuscitation trolleys and backpacks, 53 (85.5%) had one per unit/ward, 9 (14.5%) had a shared one per floor, and for 20 (32.2%) of the RRTs, they brought their one. Regarding defibrillators, 48 (77.4%) of the sample had one Automated External Defibrillator (AED) per unit/ward, 7 (11.3%) had a shared AED per floor, and 5 (8.1%) of the RRTs used their manual defibrillator.

### 3.5. The Quality Improvement Limb: How the System Measures Its Interventions and Provides Feedback to Itself and the Rest of the Organization

Regarding the measurement and registration of quality indicators, some of the most used are the following: Ulstein Registry (27.4%), number of CA (40.3%), number of ERR activations, reason for activation (30.6%), and result of the intervention (43.5%).

We asked the participants about the use of structured communication and teamwork techniques. Concerning structured communication tools, 4 (6.5%) of the centers use the SBAR tool.

Concerning the use of briefing, it was used if an adverse event happens in 4 (6.5%), sometimes in 7 (11.3%), and routinely in 7 (11.3%). About the use of huddles, in 1 (1.6%) center they are performed if an adverse event happens, and they are performed sometimes in 4 (6.5%) of them.

The most used one is clinical debriefing. This technique was used routinely in 10 (16.1%) of the centers, sometimes in 21 (33.9%), and if an adverse event happens in 8 (12.9%).

## 4. Discussion

We conducted a nationwide cross-sectional study to describe the characteristics of the process of the patient at risk of clinical deterioration outside the ICU in the Spanish context, based on the four limbs of the RRS concept [10]. This is the first study of this nature done in Spain.

In terms of the concept of SRR, we found that in our context we are at an early stage of the implementation and establishment process, as it is not yet a generalized and standardized concept in most of our hospitals. The centers that have it are still in the process of maturing the system and adapting it to their context.

Despite the ambiguous evidence in terms of the effectiveness of RRS [15,16,17,18], in Spain, between 2009 and 2010, the Department of Health and Social Policy published a report on national standards for ICUs [19] and acute hospitals wards [20], with a recommendation to implement extended critical care service following RRS principles, including the use of EWS and RRTs. From 2014 to the present day, this point has become a specific criterion for ICUs’ and wards’ quality standards assessment [19,20]. From the results obtained in our research, only 32 of the 62 hospitals have established an RRT, 15 of them straight after the national recommendations were published.

Our study is one of the few international pieces of research analyzing the full RRS concept at a nationwide level. Hospitals answered our survey from 13 out of 17 regions and two autonomous cities present in Spain. Responses were received from 62 hospitals (response rate 20.6%) of the 300 centers with ICU registered at the SEMICYUC website. Similar studies in other countries, which focused on some of the limbs of the RRS, found widely varying response rates: one in Australia (36.1% response rate) [21] and four in the United States of America (USA) [21,22,23,24], with the most recent one being published in 2018 with 107 hospitals (35% response rate); two in the Netherlands [25,26], the last published being in 2019 with 71 hospitals (92% response rate); one in Finland in 2014, with 51 hospitals (93% response rate) [27], and one in Rome in 2002 [28], where 32 centers were contacted but it is not clear if they had a 100% of response rate. The response rate obtained in the present study is above the average of previously published response rates. With the research design and the selection process of the participating hospitals, our study can give an overview of how Spanish hospitals have built their RRS.

In terms of the afferent limb, the optimum monitoring interval is unknown, but ideally should be frequent enough to detect the risk of deterioration early [10]. In our context, this is mostly done only once per shift, or in most cases, a minimum of once per shift and contemplating other frequencies, but this relies on the professional’s criteria, as shown by Ludikhuize et al. [25]. This could be a problem in terms of detecting and responding to clinical deterioration as per causes of “failure to rescue” at a system level [10]. “Track and trigger” systems are used in other contexts [21,22,25,26,27] to help professionals discriminate against instability and give clear instructions on how to proceed over time. In our context, this is not an extended practice. Most of the measured vital signs are the same as those found in other similar studies [21,25,27,29]. Despite its importance due to different factors and implications [10,30], respiratory rate is only measured in half of our sample, which is a risk to patient safety and a barrier to detecting clinical deterioration. Technology and bedside-automated transfer of vital signs to health records have an impact on mitigating human error in measuring, recording, and interpreting them [10,25], but this is not yet an extended practice in Spain. In 94% of our sample, nurses can activate directly the RRTs, in contrast with Ludikhuize et al. [25], wherein 48% of the centers’ RRTs were activated by nurses only if doctors did not respond or responded late. Our evidence correlates with Woods et al. [24]. As in other studies, the most common activating methods for the RRT found are unidirectional communication with the hospital switchboard which activates pagers [22,24,28] and/or bidirectional call with the RRT via telephone [28]. The standardized and recommended number in Europe for the activation of RRTs within hospitals is 2222. We found 56 types (between 1 and 9 digits each) of different activation numbers, with only one institution using 2222. Similar data were found in France [31] and Italy [32]. This can represent a risk and delay in activating RRTs due to actual Spanish hiring policies and migration of personnel within different hospitals in the same city or between Spanish counties.

In terms of the efferent limb, we found some differences, mostly with RRT composition, which might be associated with contextual factors within medical and nursing professions and specialties, professional development, competencies, and advanced clinical roles within countries. Regarding RRT composition, our sample was quite homogeneous, with ICU consultants and/or trainee and critical care nurses attending the calls, and with the RRT depending on ICU service. In Spain, RRTs can be attended/led by anesthetics, internal medicine, emergency medicine, and cardiology depending on the context of the hospital and local protocol. This is in contrast to the evidence found in other articles, where composition and leading roles still vary much more, and with the presence of other professionals such as respiratory therapist, nurse practitioners, nurse anesthetist, etc. [22,23,25,27,28,33]. In terms of RRT leadership and role distribution, in our study we found these two tasks to be preestablished in half of the sample, which correlates with other studies where team leadership [21,23,27] is predefined but not happening in the same way with the roles [33].

Concerning the administrative limb, and correlating it with similar evidence [10,23,25,26,27], early and regular training for involved staff in all aspects and levels of the detection and response of deteriorating patients should have a strong presence. In terms of resuscitation material within general wards, like other contexts [23], we found a wide standardization and distribution of this material in our context, in contrast with other experiences [28]. In terms of quality improvement limb, registering and analyzing RRS activity indicators has been described as difficult [24]. Similar indicators have been found in other studies [21,23], but this is still a big area of improvement in our context.

Communication and teamwork techniques used by RRTs [22,25,27], such as SBAR, briefing, huddle, and clinical debriefing, have demonstrated evidence [34,35,36,37,38,39] on different positive outcomes of teamwork and patient safety. In our context, few sample are currently using these techniques, with clinical debriefing being the most used.

In terms of limitations, as explained before, an online survey was distributed by email by SEMICYUC to all ICU medical directors with instructions to either complete it if they were the person responsible for the RRS or to deliver it to the responsible professional in charge of the RRS. Even though an effort on this matter was made, we are unable to verify the nature of the surveys due to anonymity. Furthermore, limitations related to bias of self-reported questionnaires could appear. A possible weakness could appear, even though the project was supported by SEMICYUC. The survey was voluntary, being sent and re-sent during and between COVID-19 pandemic waves, where some hospitals decreased their rapid response activity due to the high volume of work inside the ICUs. Being a descriptive online survey, we are not able to relate these findings with specific outcomes (i.e., reduced number of CAs, improved mortality, staff satisfaction, etc.). Another limitation of this research is the low response rate compared with the total number of ICUs in Spain, which might be translated into low generalizability and external validation of these findings.

## 5. Conclusions

From our findings we can conclude that in our context, RRS, as international literature and recommendations define, is a concept that is still at an early stage of development and establishment. Currently, there is a widespread heterogeneous practice within centers in the process of care of the patient at risk of clinical deterioration outside the ICU, which can benefit from the standardization of it, or at least in some parts, based on the actual international experiences and recommendations.

We can conclude that education on the importance of adequate measuring and registration of all vital signs mentioned above is key for early detection of clinical deterioration; another successful factor to implement is the standardization of a “track and trigger” system, with help of technology, when possible, but should not be a barrier to implement it early.

As a safety net, organizations must provide 24/7 professionally trained clinicians for the management of the critically ill patient, doctors, and nurses.

Standardization of human, educational, and material resources has a positive benefit on the process of the patient at risk of clinical deterioration.

Registering activity data from the RRS is vital to provide feedback to RRTs and the rest of the organization, implementing new pathways of working. The use of structured communication tools and teamwork techniques must be trained and implemented.

It is very important to remark that to make RRS’s work to its full potential, it needs to be supported from all levels of the involved stakeholders, from organizational leaders and policy-decision makers to general-specialized wards and RRT staffing.

As an important fact to remember, RRS are supported by four limbs, which all feedback and help grow each other cyclically in time.

## Figures and Tables

**Table 1 ijerph-19-12627-t001:** Afferent limb: detecting clinical deterioration.

Frequency on Measuring Vital Signs	N	%
Once every shift, contemplating other frequencies	30	48.4
Once per-shift only	27	43.5
As per nurse criteria	4	6.5
As per doctor criteria	1	1.6
Vital signs measured	N	%
Respiratory rate	31	50
Fraction of Inspired Oxygen (FiO_2_)	26	41.9
Oxygen Saturation (SpO_2_)	42	67.7
Heart rate	47	75.8
Systolic blood pressure	62	100
Temperature	62	100
Level of consciousness	27	43.5
Urine output	1	1.6
Vital signs registration	N	%
Registered in paper, then manually into electronic records	41	66.1
Automated from bedside to electronic records	17	27.4
Paper, no electronic records	4	6.5
Discriminating abnormality	N	%
None–professional criteria	43	69.4
Standardized scores to interpret and respond	12	19.4
Guidelines and/or protocols	7	11.3
Automated alarms on abnormal parameters	N	%
None	41	66.1
Vital signs	12	19.4
Laboratory-blood tests	12	19.4
Laboratory-Microbiology	9	14.5
Radiology	0	0

**Table 2 ijerph-19-12627-t002:** Afferent limb: activating the system.

Professional Able to Activate RRT	N	%
All professionals (including, HealthCare Assistants, Physiotherapist, etc.)	24	39
Only doctors and nurses	34	55
Nurses, only if responsible doctor not available	3	5
Only doctors	1	1.6
Method for activating RRT	N	%
Automated alarm directly to RRT	0	0
Emergency button at bedside	3	4.8
Telephone call–Unidirectional–No direct communication with RRT	17	27.4
Telephone call–Bidirectional–direct communication with RRT	50	80.6
Automated alarm directly to RRT	0	0
Moment of activation	N	%
Risk and/or early signs of clinical deterioration	7	11.3
Established signs of instability	50	80.6
Moments prior to cardiac arrest	49	79
Established cardiac arrest	50	80.6
Changes in vital signs	53	85.5
Signs of clinical deterioration	51	82.3
Concern from the responsible professional	35	56.5
Syndromic presentation	25	40.3
Laboratory/blood test abnormalities	23	37.1
Blood test abnormalities without changes in vital signs	17	27.4
Changes in vital signs	53	85.5

**Table 3 ijerph-19-12627-t003:** Efferent limb: characteristics.

**RRT Most Common Responders**	**N**	**%**
Only on call physician–intensivist	17	15%
One RRT	46	74%
More than one RRT	16	26%
**RRT Most Common Members**	**N**	**%**
Intensive Care Consultant	59	95.2
Intensive Care Resident	33	53.2
Intensive Care Nurse	15	24.2
RRT full time dedicated Nurse	2	3.2
Anesthesia Consultant	9	14.5
Anesthesia Resident	5	8.1
Emergency Department Consultant	4	6.5
Emergency Department Nurse	1	1.6
Cardiology Consultant	3	4.8
Cardiology Resident	3	4.8
General/Internal Medicine Consultant	4	6.5
General/Internal Medicine resident	4	6.5
Other specialties Consultants	2	3.2
No Resident	3	4.8
Nurse from other specialties	2	3.2
No nurse	8	12.9
Porter	7	10.6
Healthcare Assistant	2	3%
Charge Nurse/Supervisor	1	1.6
RRT Availability	N	%
24/7 h	57	91.1
Monday to Friday: 8–15 h or 8–17 h)	3	4.8
Everyday 8–22 h	1	1.6
Monday to Friday 8–20 h (10–17 h dedicated nurse)	1	1.6
Team Leadership	N	%
Preestablished	57	92.8
Established in-situ	5	7.2
Role distribution	53	85.5
Preestablished	42	67.7
Established in-situ	20	32.3

**Table 4 ijerph-19-12627-t004:** Administrative limb: educational resources.

Resuscitation Guidelines	N	%
ERC	44	71
AHA	3	4.8
ERC/AHA depending on unit	7	11.3
ERC/AHA depending on Team Leader	8	12.9
Ward nurses training	N	%
No standardized training plan	12	19.4
Who and how to call in an emergency	25	40.3
Basic Life Support (BLS)	34	54.8
Immediate Life Support (ILSI	24	38.7
Advanced Life Support (ALS)	24	38.7
RRT members training	N	%
No standardized training plan	11	17.7
Basic Life Support (BLS)	14	22.6
Immediate Life Support (ILSI	6	9.7
Advanced Life Support (ALS)	26	41.9
Other Specific training/experience	45	72.6
Simulation for training	53	85.5
No simulation activities	16	25.8
Some simulation activities	32	51.6
Trained simulation instructors	25	40.3
Dedicated space for sim training	7	11.3
Dedicated center/unit and professionals’ team	10	16.1
Depends on a University	1	1.6
Opening a center in the next 1 year	1	1.6

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
