# Peer review of "National Survey: How Do We Approach the Patient at Risk of Clinical Deterioration outside the ICU in the Spanish Context?"

_ijerph, 2022, doi:10.3390/ijerph191912627_

Round 1
Reviewer 1 Report
This manuscript describes a survey of the current state of RRT in Spain, and I have some comments to the author.
Comment 1. How many hospitals in the list of hospitals with Intensive Care Units service did you send the questionnaire to and did you receive responses from 62? (In other words, what was the recovery rate of questionnaire?)
Comment 2. I think Table1 shows the provinces in Spain, is this table necessary? I don't think this table makes sense, because the names of the provinces or provinces are completely incomprehensible to people outside of Spain. I think that this content should be described in the text in the text (I don't think it is necessary to go as far as the detailed percentage by region).
Comment 3. What is the abbreviation for “HCA’s” in All professionals in Table 3? If using an abbreviation, please indicate it by adding a footnote to the table.
Comment 4. Table 5 and its associated text also make similar points to the comments above. For ERC and AHA, please show full spelling when first appearing. Please indicate the full spelling of the abbreviation ERR as well.
Comment 5. In relation to the above comment, if AHA means the American Heart Association and what is shown in Table 5 is the AHA guideline, then I think it should be cited as literature. The same is true for ERCs.
Comment 6. References are not in the format required by the journal.
Comment 7. What does the sentence "(This is the abstract section, maximum 200 words in total. Please use the subheadings 55 given.)" at the end of the abstract mean?
Author Response
IJERPH - MPDPI
Manuscript ID IJERPH -1939440 entitled " National Survey: How Do We Approach the Patient at Risk of Clinical Deterioration outside the ICU in the Spanish Context?”
Dear Editor,
On behalf of the authors, I would like to thank you for giving us the opportunity to submit another version of our manuscript. Furthermore, we would like to thank the reviewers for their suggestions. We have made the changes accordingly covering all suggestions and comments in the revised version of the manuscript, achieving a clearer article.
REVIEWER 1
COMMENTS:
- Comment 1. How many hospitals in the list of hospitals with Intensive Care Units service did you send the questionnaire to, and did you receive responses from 62? (In other words, what was the recovery rate of questionnaire?):
Thank you for the recommendation. We have changed it.
Answer:
Page 4, lines 158-159. “Responses were received from 62 hospitals (20,6%) of the 300 centers with ICU registered at the SEMICYUC website”.
Page 9, lines 262-263. “Responses were received from 62 hospitals (response rate 20,6%) of the 300 centers with ICU registered at the SEMICYUC website”.
- Comment 2. I think Table1 shows the provinces in Spain, is this table necessary? I don't think this table makes sense, because the names of the provinces or provinces are completely incomprehensible to people outside of Spain. I think that this content should be described in the text in the text (I don't think it is necessary to go as far as the detailed percentage by region).
Thank you for the recommendation. We have changed it.
Answer: We have deleted the table and make a description in the text as suggested.
Page 4, lines 160-162. “Responses were distributed by 13 of the 17 counties in Spain. We didn´t obtained any answers from the following 4 regions: Castilla la Mancha, Comunidad Foral de Navarra, Extremadura y La Rioja. Neither as from the autonomous cities of Ceuta and Melilla”.
- Comment 3. What is the abbreviation for “HCA’s” in All professionals in Table 3? If using an abbreviation, please indicate it by adding a footnote to the table.
Thank you for the recommendation. We have changed it.
Answer: We have written the no abbreviated form, “Health Care Assistant”.
Page 6, table 2 (table 3 before deleting the “responses by county” table).
- Comment 4. Table 5 and its associated text also make similar points to the comments above. For ERC and AHA, please show full spelling when first appearing. Please indicate the full spelling of the abbreviation ERR as well.
Thank you for the recommendation. We have changed it.
Answer: We have written the no abbreviated form.
Page 7, lines 210-211. “…used are the ones established by European Resuscitation Council (ERC) (71%), other guidelines are used, like American Heart Association (AHA), relying on the decission…”.
Regarding “ERR “(Spanish version of RRT), RRT non abbreviated form is described in the abstract section, page 1, line 40.
- Comment 5. In relation to the above comment, if AHA means the American Heart Association and what is shown in Table 5 is the AHA guideline, then I think it should be cited as literature. The same is true for ERCs.
Thank you for the recommendation. We have changed it.
Answer: We have added as recommended.
- Comment 6. References are not in the format required by the journal.
Thank you for the recommendation. We have changed it. Pages 12 – 13.
Answer: We have changed it into Chicago style as required by the journal.
- Comment 7. What does the sentence "(This is the abstract section, maximum 200 words in total. Please use the subheadings 55 given.)" at the end of the abstract mean?
Thank you for the recommendation. We have changed it.
Answer: Deleted as it was a mistake from the draft. Page 1, lines 55-56.
Reviewer 2 Report
The authors have undertaken an investigation to characterize the use of Rapid Response Systems nationally in Spain. Such studies are not novel, but are uncommon and important. This is because they provide a benchmark for those within and outside the country. It provides baseline data to identify deficiencies and target remediation projects. It also can highlight strengths for others to emulate both within and outside the country. I have just one concern and that relates to the results. The investigators distributed a survey and laudably they received over 60 replies. However, the reader does not know the sampling rate because the number of hospitals in the country is not reported and even more importantly, the number of surveys mailed is also excluded. This is essential data for the reader to determine the degree to which the survey distribution represents the country, and the survey results represents those surveyed. I think this is essential data without which the manuscript suffers significantly.
Author Response
IJERPH - MPDPI
Manuscript ID IJERPH -1939440 entitled " National Survey: How Do We Approach the Patient at Risk of Clinical Deterioration outside the ICU in the Spanish Context?”
Dear Editor,
On behalf of the authors, I would like to thank you for giving us the opportunity to submit another version of our manuscript. Furthermore, we would like to thank the reviewers for their suggestions. We have made the changes accordingly covering all suggestions and comments in the revised version of the manuscript, achieving a clearer article.
REVIEWER 2
COMMENTS:
- Comment 1. How many hospitals in the list of hospitals with Intensive Care Units service did you send the questionnaire to, and did you receive responses from 62? (In other words, what was the recovery rate of questionnaire?):
Thank you for the recommendation. We have changed it.
Answer:
Page 4, lines 158-159. “Responses were received from 62 hospitals (20,6%) of the 300 centers with ICU registered at the SEMICYUC website”.
Page 9, lines 262-263. “Responses were received from 62 hospitals (response rate 20,6%) of the 300 centers with ICU registered at the SEMICYUC website”.
Round 2
Reviewer 1 Report
Thank you for responding to my comment and correcting the manuscript.